Giant, swimming mouths: oral dimensions of extant sharks do not accurately predict body size in Dunkleosteus terrelli (Placodermi: Arthrodira)

Engelman Russell neovenatoridae@gmail.com
Department of Biology, Case Western Reserve University , Cleveland , OH , United States of America
Farke Andrew
Electronic publication date: 2023 Apr 10
Publication date: 2023
Volume: 11
Electronic Location ID: e15131
Received 2022 Dec 15; Accepted 2023 Mar 6
Copyright: ©2023 Engelman
Copyright year: 2023
Copyright holder: Engelman
License: This is an open access article distributed under the terms of the Creative Commons Attribution License, which permits unrestricted use, distribution, reproduction and adaptation in any medium and for any purpose provided that it is properly attributed. For attribution, the original author(s), title, publication source (PeerJ) and either DOI or URL of the article must be cited.
License URL: https://creativecommons.org/licenses/by/4.0/

Keywords: Devonian, Ichthyology, Allometry, Paleozoic, Diet, Body size, Size estimation, Predator-prey size

Funding: The authors received no funding for this work.

==============================
Background

The size of Dunkleosteus and other late Devonian arthrodire placoderms has been a persistent problem in paleontology. The bony head and thoracic armor of these animals are typically the only elements preserved in the fossil record, with the rest of the body being lost during fossilization. Accurate length estimates of arthrodires are critical for reconstructing the paleobiology of these taxa and Devonian paleoecology more generally. Lengths of 5.3–8.8 m were proposed for Dunkleosteus based on allometric relationships between upper jaw perimeter and total length in extant large-bodied sharks. However, these methods were not statistically evaluated to determine if allometric relationships between body size and mouth size in sharks reliably predicted size in arthrodires. Several smaller arthrodire taxa are known from relatively complete remains, and can be used as independent case studies to test the accuracy of these methods.

Results

Length estimates for Dunkleosteus are evaluated through an examination of mouth proportions in complete arthrodires and fishes more generally. Currently accepted lengths of 5.3–8.8 m for D. terrelli are mathematically and biologically unlikely for three major reasons: (1) Arthrodires have larger mouths than sharks at similar body sizes. (2) upper jaw perimeter and mouth width produce extreme overestimates of body size (at least twice the actual value) in arthrodires known from complete remains. (3) Reconstructing Dunkleosteus using lengths predicted by upper jaw perimeter results in highly unusual body proportions, including extremely small, shrunken heads and hyper-anguilliform body plans, not seen in complete arthrodires or fishes more generally.

Conclusions

Length estimates for arthrodires based on the mouth dimensions of extant sharks are not reliable. Arthrodires have proportionally larger mouths than sharks, more similar to catfishes (Siluriformes). The disproportionately large mouths of arthrodires suggest these animals may have consumed larger prey relative to their body size than extant macropredatory sharks, and thus the paleobiology and paleoecology of these two groups may not have been exactly analogous within their respective ecosystems.

Introduction

Dunkleosteus terrelli and other giant late Devonian arthrodires have fascinated people for more than a hundred years (Newberry, 1875; Newberry, 1889). These extinct fishes have attracted attention from both specialists and laypersons alike due to their bony armor, bladed mouthparts, and large size, with these taxa representing some of the “first true apex predators seen in the vertebrate fossil record” (Anderson & Westneat, 2007; Anderson & Westneat, 2009; Long, 2010: p. 56; Young, 2003). However, although Dunkleosteus and other, similar arthrodires were clearly very large animals, determining their exact size has been challenging. This is primarily because these fishes combine an ossified dermal armor covering the front third to half of the animal with a mostly cartilaginous endoskeleton (Denison, 1978; Heintz, 1932). Thus, only the bony head and thoracic armor of most arthrodires are typically preserved in the fossil record, with the back half of the animal being lost during fossilization. This makes it difficult to determine exactly how much of the animal is missing, and how large arthrodires like Dunkleosteus really were.

Determining the size of Dunkleosteus and other large arthrodires has broader evolutionary and paleoecological implications beyond the sensationalism of “what species is the biggest sea monster”. Almost every aspect of an organism’s biology is influenced to some degree by body size (Schmidt-Nielsen, 1984), and thus accurate estimates of size are crucial for reconstructing the paleobiology and paleoecology of extinct animals. For example, the size of prey items eaten by a predator is strongly correlated with both animals’ body size, due to gape limitations, energetic costs versus benefits, and the ability of a given predator to overpower, process, and consume a given prey item (Brose, 2010; Costa, 2009; Scharf, Juanes & Rountree, 2000). Thus, as apex predators in many Devonian ecosystems (Hussakof, 1906; Williams, 1990), understanding how large Dunkleosteus and similar arthrodires grew is critical for reconstructing Devonian paleoecology (e.g., as in Chevrinais, Jacquet & Cloutier, 2017). This is specifically important in terms of identifying the types of animals large arthrodires might have preyed on and what, if anything, might have preyed on them (Hall, Ryan & Scott, 2016; Williams, 1990).

Accurate estimations of arthrodire body size are also important in understanding broader trends in vertebrate evolution. More specifically, the Devonian is considered to mark a major period of body size diversification in gnathostomes (Choo et al., 2014; Dahl et al., 2010; Lamsdell & Braddy, 2010; Sallan & Galimberti, 2015). Prior to the Devonian, most vertebrates measured less than 35 cm in length (with the largest being ∼1 m; Choo et al., 2014) but by the end of the period appear to display body size distributions comparable to modern marine faunas (Sallan & Galimberti, 2015). Several abiotic and biotic drivers have been proposed for this expansion in body size, including increases in atmospheric oxygen (Choo et al., 2014; Dahl et al., 2010), and competitive release from eurypterid arthropods (Lamsdell & Braddy, 2010). However, determining when and how this diversification in body size occurred (and therefore whether its timing correlates with any particular biotic or abiotic event) is heavily dependent on accurate estimates of body size for the vertebrates that lived during this critical period. As the largest animals in their environments, Dunkleosteus and other large arthrodires are critical for setting maximal limits on vertebrate size during the Devonian (Sallan & Galimberti, 2015), and therefore constraining the mode and tempo of this important evolutionary event.

In spite of the broad interest in and scientific importance of the body size of Dunkleosteus terrelli, most previous estimates of this species’ size have been based on no rigorous quantification (see discussion in Ferrón, Martinez-Perez & Botella, 2017). Most previous studies discussing the size of Dunkleosteus simply offer speculative estimates of body length without describing how these values were calculated or what anatomical data were used to produce them (e.g., Anderson & Westneat, 2007; Anderson & Westneat, 2009; Carr, 2010; Denison, 1978; Romer, 1966). More recently, Ferrón, Martinez-Perez & Botella (2017) set out to rectify this lack of rigorous quantification, attempting to estimate the body size of Dunkleosteus using a variable these authors predicted would universally correlate with body size across fishes: mouth perimeter. The rationale behind this is the size of prey a fish commonly takes is strongly constrained by mouth size. This can be either directly, as in most gape-limited teleosts, or indirectly, as in sharks and other “dismemberment predators” (barracuda, piranha, etc.). “Dismemberment predators” are able to circumvent gape limitations and eat prey items larger than their mouth by tearing them into consumable pieces (Grubich, Rice & Westneat, 2008; Helfman & Clark, 1986). However, prey size still plays an important role in prey choice for these predators as it correlates to handling time and the organism’s ability to overpower, process, and consume the prey item (Bethea et al., 2006; Juanes & Conover, 1994; Lowry et al., 2009; Lucifora et al., 2008; Scharf et al., 1997). Thus, mouth size and body size are strongly correlated in most predatory fishes (Karpouzi & Stergiou, 2003; Scharf, Juanes & Rountree, 2000).

Ferrón, Martinez-Perez & Botella (2017) estimated body size in Dunkleosteus terrelli using upper jaw perimeter (UJP) and the dataset of Lowry et al. (2009), who collected UJP and total length in a series of large modern sharks (sphyrnids, carcharhinids, and lamnids) for use in forensic shark bite analysis. Ferrón, Martinez-Perez & Botella (2017) concluded D. terrelli was larger than previously suggested (though not as large as the most extreme prior estimates of 10 m), with subadult to adult individuals of D. terrelli estimated as measuring 5.3–6.9 m long and the largest known specimen of Dunkleosteus estimated as 8.79 m in total length. However, at the same time these methods were not statistically evaluated to determine if they produced reliable results in arthrodires. Additionally, Ferrón, Martinez-Perez & Botella (2017) did not demonstrate sharks and arthrodires were similar enough in body proportions to make the former a reasonable anatomical model for the latter. This is important given arthrodires differ from sharks in having bladed mouthparts restricted to the anterior half of the jaw (whereas the tooth-bearing regions of modern shark jaws span most of the jaw cartilages) and the body proportions of placoderms have never been evaluated in a comparative context (except for Amazichthys; Jobbins et al., 2022). This is a particular concern given Lowry et al. (2009) conducted their study on a taxonomically restricted dataset of lamnid and carcharhinid + sphyrnid sharks (i.e., the species of concern in forensic shark bite analysis), and thus did not consider the entire extant diversity of large-bodied, macropredatory sharks (e.g., hexanchiids, somniosids) in their model.

Some arthrodire species are known from complete remains and/or body outlines (e.g., Coccosteus cuspidatus). However, most of these arthrodires are much smaller than Dunkleosteus terrelli, the largest being less than 1 m in length (Jobbins et al., 2022). As noted by Ferrón, Martinez-Perez & Botella (2017), over-relying on comparisons with these smaller arthrodires when reconstructing Dunkleosteus is potentially fraught with peril, especially as Dunkleosteus is an order of magnitude larger from these species and strongly differs from them in its inferred paleoecology (Carr, 2010). However, these arthrodires provide a useful case study for testing whether mouth dimensions represent a reliable proxy for estimating total length in arthrodires, given the anatomy of these species more closely resembles D. terrelli than sharks. In this study, I critically evaluate the methodology and length estimates of Ferrón, Martinez-Perez & Botella (2017), using both the dataset of Lowry et al. (2009) and a wider dataset of extant fishes.

Materials and Methods

Measurements of Dunkleosteus terrelli and definition of measurements

This study primarily focuses on four measurements: upper jaw perimeter (UJP), mouth width, mouth length, and head length (Figs. 1–2). UJP is the measurement used by Ferrón, Martinez-Perez & Botella (2017) to estimate length in Dunkleosteus terrelli. Mouth width was chosen as a proxy for UJP in broader analyses of fishes. Direct measurements of UJP are only available for a very narrow dataset of large-bodied sharks (see below), but mouth width data is easily measurable and can be more easily compared across a broader range of fishes. Mouth length is used primarily for anatomical comparisons between taxa in this study, rather than evaluating size estimates, given it is much more interspecifically variable. Finally, head length was chosen as an independent test of estimates based on mouth dimensions. Although head proportions are variable across fishes, if estimates using mouth and non-mouth variables were at least somewhat similar it would suggest these length estimates were accurate.

Figure 1 Definitions of mouth measurements used in present study.

Dunkleosteus terrelli (CMNH 6090) in anterior (A) and right lateral (B) view, showing how upper jaw perimeter (UJP), mouth length, and mouth width were measured in this study for arthrodires. Inner mouth width denoted by dashed lines.

Figure 2 Figure showing how head length was defined in various fishes.

(A) an osteichthyan with an opercular cover (Micropterus dolomieu, CMNH teaching collection), (B) an elasmobranch chondrichthyan with multiple gill slits (Carcharhinus obscurus; proportions from Garrick, 1982), (C) an arthrodire (Coccosteus cuspidatus; modified from Miles & Westoll, 1968), and (D) Dunkleosteus terrelli (CMNH 6090). Dashed line in D shows how the length of the skull roof to the cranio-thoracic joint is equal to the length of the skull to the end of the branchial cavity.

Measurement data for Dunkleosteus terrelli was gathered from the four mounted specimens of this species at the Cleveland Museum of Natural History measured by Ferrón, Martinez-Perez & Botella (2017): CMNH 7424, CMNH 6090, CMNH 7054, and CMNH 5768. Upper jaw perimeter was measured using the same criterion as Ferrón, Martinez-Perez & Botella (2017: fig. 1c), measuring along the ventral margins of the suborbital plates from the termination of the supraoral sensory line along the perimeter of the mouth, making a small detour to encompass the border of the supragnathal plates. Although the present author was able to measure upper jaw perimeter in specimens of Dunkleosteus terrelli firsthand, these values were close enough to those of Ferrón, Martinez-Perez & Botella (2017) (e.g., UJP = 116 cm in CMNH 5,768 by the present author versus 120 cm in Ferrón, Martinez-Perez & Botella (2017)) that the measurements of Ferrón, Martinez-Perez & Botella (2017) were used to ensure maximum comparability with the previous study.

Mouth width in arthrodires was measured as the mediolateral length between the ventral ends of the supraoral sensory line on the suborbital plate (Fig. 1). This definition was used to match the landmarks Ferrón, Martinez-Perez & Botella (2017) used to define the terminal ends of UJP, in order to maximize comparability between the measurements. Additionally, this definition of mouth width is very similar to its definition in extant sharks (MOW of Compagno, 1984), which is measured across the outer edges of the mouth in external view. However, it is possible that this might be overestimating the width of the jaw due to the presence of soft tissues in life. To account for this a second measurement of inner mouth width, measured between the inner angles of gnathal plates at the point where the inferognathal intersects with the suborbital in lateral view, was also considered (though it could only be measured for Coccosteus and Plourdosteus among complete arthrodires). These represent maximum and minimum possible mouth widths in arthrodires.

There are other reasons to believe external mouth width might be more reasonable than internal mouth width when comparing sharks and arthrodires. In sharks, mouth width (sensu Compagno, 1984) is measured across the maximum external width of the mouth in the living animal, rather than internally across the jaw elements. In some chondrichthyan taxa with pronounced labial folds (e.g., hexanchiids) the external mouth width can be significantly larger than mouth width measured across the bony/cartilaginous jaw elements (S Adnet and F Mollett, pers. comm., 2022). Indeed, the condition in hexanchiids is actually very similar to what is seen in Dunkleosteus, in which the maximum width of the mouth (as defined by external measurements/the supraoral lateral line) is actually greater than the internal jaw width defined by the jaw articulation. Thus, the landmarks used by Ferrón, Martinez-Perez & Botella (2017) to define the ends of the mouth are actually closer to mouth width as defined in living chondrichthyans compared to how it would be defined in most paleontological studies.

Mouth length in arthrodires was measured from the anterior end of the gnathals to the posteriormost point of the supraoral sensory canal in lateral view. It is important to specify lateral view here, as the skulls and armors of most arthrodires are mediolaterally wide and thus point-to-point measurements significantly distort the size of the animal. In reconstructions and three-dimensional arthrodire specimens (e.g., Miles & Dennis, 1979; Miles & Westoll, 1968; mounted specimens of Dunkleosteus at the CMNH) the end of the supraoral sensory canal is closely correlated with the angle of the jaws (based on the articulation of the gnathal plates), suggesting it is related to the soft tissue angles of the jaws in life. For sharks, measurements of mouth length follow Compagno (1984), in which mouth length is measured from the symphysis of the Meckel’s cartilages to the angles of the jaw along the anteroposterior axis. In osteichthyans, mouth length was measured as “upper jaw length”, i.e., the length of the maxilla and/or premaxilla in lateral view (Hubbs, Lagler & Smith, 2004).

Head length in arthrodires was measured as the anteroposterior length from the tip of the rostral plate to the cranio-thoracic joint (Figs. 2C–2D). In most fishes, head length is typically measured from the tip of the rostrum to the end of the gill chamber, though the definition of the latter landmark can vary based on the presence of a gill cover. In bony fishes and chimaeroids the end of the gill chamber is measured to the end of the operculum (Fig. 2A; Hubbs, Lagler & Smith, 2004), whereas in sharks and lampreys this value is measured to the terminal gill opening (Fig. 2B; Compagno, 1984). Measuring from the tip of the snout to the cranio-thoracic joint in arthrodires encompasses the same region included in definitions of head length in other fishes (i.e., snout, jaws, neurocranium, and gill chamber). In arthrodires, it is necessary to measure to the end of the cranio-thoracic joint as in many coccosteomorphs the neurocranium actually extends posterior to the gill chamber (Fig. 2C). By contrast, in Dunkleosteus the cranio-thoracic joint and posterior boundary of the gill chamber are located at the same anteroposterior level (Fig. 2D), which means the measured length to the cranio-thoracic joint is equivalent to the length to the end of the gill chamber. Thus, head length in Dunkleosteus will be the same regardless of whether it is measured to the cranio-thoracic joint or end of the gill chamber, and thus the results using head length are unlikely to be due to an improper anatomical definition of this term.

Comparative data in arthrodires

Comparative data was gathered for a number of smaller arthrodire taxa for which complete remains are known. These data were either collected from the previously published literature or photographs of specimens housed at the Field Museum of Natural History (FMNH), Musée d’Histoire Naturelle de Miguasha (MNHM), National Museum of Scotland (NMS), and Royal Ontario Museum (ROM). However, due to variable preservation of the head and post-thoracic anatomy, many specimens and taxa could only be included in some of the analyses performed in this study.

In some cases arthrodire measurements had to be drawn from composites of multiple individuals. This is because UJP and mouth width can often only be measured in three-dimensionally preserved specimens or life reconstructions. However, mounting an arthrodire in life position involves extracting the plates from the specimen, and thus separating them from the axial skeleton (Miles, 1970; Vézina, 1988). The post-thoracic axial skeleton of most arthrodires consists of neural and haemal arches surrounding an (unpreserved) notochord (Johanson et al., 2019; Van Mesdag et al., 2020). Thus, it cannot be easily mounted and is often left in the matrix. In other cases, parts of the axial skeleton may not be preserved, even if the mouth is measurable (i.e., part of the tail may be missing). This necessitates the use of composite specimens and reconstructions in the present study in order to compare mouth proportions between arthrodires and other fishes. However, in cases where composite specimens were used it was ensured that all proportions were comparable by scaling all material to the same body size.

Head length but no mouth measurements were available for several other complete arthrodire taxa such as Africanaspis doryssa (see Gess & Trinajstic, 2017) and Dickosteus threplandi, due to the way these specimens were preserved. These taxa were included in the analyses examining head length, but could not be included in any analyses of mouth dimensions.

Comparative dataset of extant fishes

A comparative dataset of extant fishes was generated from a sample of 3,168 extant fish observations representing 985 species, including 782 observations of sharks representing 180 species. These observations were drawn from the previously published literature, as well as from specimens measured directly at the Cleveland Museum of Natural History (CMNH), Florida Fish and Wildlife Research Institute (FSBC), and Ohio State University Museum of Biodiversity (OSU). A complete list of these data can be found in File S1. Observations primarily represent individual specimens, but in some cases literature data represent sample means as this is how morphometric data is frequently reported in the ichthyological literature. All body length data was taken as total length, with the caudal fin in as close to natural position as possible, rather than standard length (= precaudal length, which is how body length is normally reported in actinopterygians; Howe, 2002; Hubbs, Lagler & Smith, 2004). This is because total length is the measurement of choice for studies on extant sharks (Compagno, 1984) and sarcopterygians (De Vos & Oyugi, 2002), and is the more typical method of reporting length in paleontological studies.

Ferrón, Martinez-Perez & Botella (2017) estimated body length in arthrodires based on the dataset of Lowry et al. (2009) (see File S2), which contained data from 14 large, extant shark taxa (Carcharodon carcharias, Isurus oxyrinchus, I. paucus, Carcharhinus acronotus, C. brevipinna, C. falciformis, C. leucas, C. limbatus, C. obscurus, C. plumbeus, Negaprion brevirostris, Galeocerdo cuvier, Sphyrna lewini, and S. mokarran). Even the smallest individual in this dataset (68.5 cm total length) is much larger than all but the largest complete arthrodires considered here, which might be expected to cause extrapolation errors when predicting the length of smaller taxa (Schmidt-Nielsen, 1984). To compensate for this, a regression equation was created using mouth width and total length in sharks using data from the previously published literature, as mouth width measurements are available for smaller sharks such as dogfishes (Squalidae). Mouth width and UJP are expected to be highly correlated as both are measuring from similar landmarks. Estimates from the two methods were compared to test whether they produce similar results.

In contrast to sharks, mouth width was only available for a limited subset of osteichthyans, as most morphometric studies on Osteichthyes do not report this measurement. Mouth width data in this study includes 639 observations of elasmobranchs representing 166 different species and 417 observations of bony fishes from 172 different species (File S1). UJP is typically not reported for sharks in the literature, and so most of this data lacks rigorous measurements for UJP. Five dried shark jaws with known total length (two Galeocerdo cuvieri, three Carcharhinus leucas) were measured in the FSBC collections to test the accuracy of the dataset of Lowry et al. (2009) and to determine if mouth width and UJP produced comparable results in sharks (File S3).

An alternate possibility is using mouth length and mouth width to estimate UJP in extant sharks. This can be done by treating mouth width and length as the major and minor axes of an ellipse, respectively, calculating the perimeter and then dividing it in half. There is no precise mathematical solution for the calculation of the perimeter of an ellipse given its major and minor axes, but this value can be approximated using Ramanujan’s approximation for the perimeter of an ellipse (Ramanujan, 1962; Villarino, 2008). UJP was estimated for shark taxa in which mouth length and width were reported. This relationship was used to predict the length of Coccosteus cuspidatus and Dunkleosteus terrelli (using measured UJP, as Ramanujan’s approximation did not work on arthrodires, see File S4) and the results were compared to that using the dataset of Lowry et al. (2009).

However, the very largest sharks (Rhincodon, Megachasma, and Cetorhinus) are filter-feeders and thus might be expected to have larger mouths compared to predatory sharks, and many other shark groups (e.g., Squaliformes, Orectolobiformes) have anteroposteriorly narrow mouths in external view that might make comparisons with arthrodires unreasonable. Therefore, an alternate model was also run only considering macropredatory sharks that do not have large labial folds or cartilages (Carcharhiniformes, Lamniformes, and Hexanchiiformes). There is currently no evidence for extensive labial cartilages in arthrodires (J Long, pers. comm., 2023). Although Hexanchiiformes have labial folds, unlike other shark groups the labial folds do not restrict the dimensions of the mouth in external view. Considering only Carcharhiniformes, Lamniformes, and Hexanchiiformes greatly restricts the number of observations in this dataset. However, this dataset still contains a large number of sharks close to or spanning the size range of Coccosteus (Rhizoprionodon spp., Scoliodon spp., Triakidae, Scyliorhinidae).

Head length was measured in extant fishes as noted above (Fig. 2), at least in cases where data was measured firsthand. For billfishes (Istiophoriformes), paddlefishes (Polyodontidae), and halfbeaks (Hemirhamphidae), bill length (i.e., the protrusion of the upper or lower jaw beyond the mandible) was subtracted from total and head length in order to make more reliable comparisons with other fishes. Including the bill in head and total length resulted in head size being exaggerated relative to body size (head length with bill is about ∼33% total length in billfishes). Head-body proportions of Xiphactinus audax were taken from USNM V21375 from a published photo by the Smithsonian Institute (https://collections.si.edu/search/detail/edanmdm:nmnhpaleobiology_3423486) as an additional point of comparison (see below). Similarly, mouth dimensions and body length were measured in three specimens of Cladoselache (AMNH FF 240, CMNH 5047, and CMNH 5934) to test if mouth width accurately predicted body length in this extinct chondrichthyan.

Analyses

All analyses and statistical calculations were performed in R 4.2.1 (R Core Team, 2020). A knitted .html document showing the direct result of all analyses performed in this study can be found in File S4, and the original R code can be downloaded from this document as an .rmd file and rerun to replicate all analyses in this study. Most analyses were performed by simple linear regression, either untransformed or with log-transformation, and thus no additional R packages were used for data analysis, though some such as the tidyverse suite (Wickham et al., 2019) and the packages broom (Robinson, Hayes & Couch, 2022), cowplot (Wilke, 2020), ggstar (Xu, 2022), ggtext (Wilke & Wiernik, 2022), kableExtra (Zhu, 2021), magick (Ooms, 2021), magrittr (Bache & Wickham, 2022), readxl (Wickham & Bryan, 2022), and scales (Wickham & Seidel, 2022), were used for data visualization. Accuracy statistics were calculated using methods described in Engelman (2022b).

In addition to the statistical analyses, drawings were made of CMNH 5768, the largest complete individual of Dunkleosteus terrelli and CMNH 6090, a smaller individual of D. terrelli. These drawings were made from direct observations of the specimens at the Cleveland Museum of Natural History as well as 3D surface scans available via University of Michigan UMORF (https://umorf.ummp.lsa.umich.edu/wp/specimen-data/?Model_ID=1336) and MorphoSource (morphosource.org). Thus, the dimensions of the bony armor in this figure represent the actual dimensions of CMNH 5768 and CMNH 6090 as preserved, rather than approximations. These specimens were mostly depicted as is, the only adjustment being the ventral armor was restored as curved based on the estimated curvature of the body (see File S4 for reasoning).

Results

Upper jaw perimeter and mouth width

Coccosteus cuspidatus is estimated to be 100.8 cm using UJP (Table 1), nearly 2.5 times the actual length of this taxon. The actual total length of 39.4 cm is within the 95% confidence intervals of the UJP equation (30.9–170.7 cm), but only barely. However, because Coccosteus is significantly smaller than the smallest shark in the dataset of Lowry et al. (2009) (a 68 cm long individual of Carcharhinus limbatus) it could be argued this inaccuracy is the result of extrapolation, which has been known to produce significant errors in other attempts to estimate the size of organisms (see Millien, 2008; Schmidt-Nielsen, 1984). This is supported by the fact UJP significantly overestimates the length of a 64 cm neonatal bull shark (FSBC 18083; File S4: table 5.2), though this result could also be due to allometric variation in mouth size across ontogeny. On the other hand, the UJP of Coccoseus (9.78 cm) is within the range of values reported by Lowry et al. (2009) (8.5 to 99 cm), so in theory extrapolation should not be an issue.

Table 1 Total length estimates in Coccosteus cuspidatus and Dunkleosteus terrelli using mouth width and upper jaw perimeter.

UJP-based estimates calculated using methodology of Ferrón, Martinez-Perez & Botella (2017) and dataset of Lowry et al. (2009). % Dif represents the percent difference between these two estimates. All measurements in cm.

Specimen	Taxon	Mouth width	Estimated total length	
			Using mouth width	Using UJP	% Dif.	
N/A	Coccosteus cuspidatus	6.87	89.1 (58.8–135.0)	100.8 (30.9–170.7)	13%	
CMNH 7424	Dunkleosteus terrelli	30.0	296.4 (195.5–449.5)	321 (251–390)	8%	
CMNH 6090	Dunkleosteus terrelli	51.4	377.9 (249.1–573.4)	534 (462–605)	16%	
CMNH 7124	Dunkleosteus terrelli	52.0	463.3 (305.9–704.8)	576 (504–648)	24%	
CMNH 5768	Dunkleosteus terrelli	71.6	603.0 (397.0–915.8)	688 (615–762)	14%	

When applied to five sharks of known total length (File S3), UJP and mouth width produced estimates that agreed with one another. In the four sharks which spanned the range of lengths examined by Lowry et al. (2009) estimated lengths were close to the actual value. A fifth shark, the aforementioned neonatal Carcharhinus leucas, had its length more accurately estimated under mouth width than under UJP (File S4: table 5.2), which is expected given the latter model spans a broader range of body sizes. This result suggests UJP and mouth width are capturing the same morphometric signal, and thus mouth width can be used as a substitute to evaluate the relationship between mouth dimensions and body size in sharks and arthrodires.

Log mouth width correlates strongly with log total length in extant elasmobranchs (Fig. 3), regardless of whether individual specimens (r2 = 0.909, percent prediction error [%PE] = 16.75) or species means (r2 = 0.890, percent prediction error [%PE] = 17.15) are considered. Mouth width is positively allometric with respect to total length in sharks (slope = 1.115 ± 0.010 in the specimen-level equation when total length is made the predictor variable; File S4: section 5.1). This pattern remains even when considering species averages excluding sexually immature specimens (slope = 1.105 ± 0.031 when total length is the predictor variable; see File S4: section 5.2). Mouth width of Cladoselache sp. produced estimated body lengths close to the actual values for these specimens (black rectangles in Fig. 3). Thus, mouth width appears to be an accurate predictor of body size within Chondrichthyes.

Figure 3 Plot of mouth width versus total length in sharks (rectangles) and arthrodires (stars) on a log10 scale.

Black stars represents arthrodires with known total lengths and mouth widths, whereas yellow stars represent the estimated lengths of Dunkleosteus terrelli from Ferrón, Martinez-Perez & Botella (2017) compared with measured mouth widths of the same specimens. Magenta stars are specimens of Amazichthys trinajsticae and an undescribed aspinothoracid at the CMNH in which the mouth width cannot be directly measured, but can be approximated based on skull roof width. Orange squares represent the chondrichthyan genera Planonasus, Cephaloscyllium, and Isistius, which have unusually wide mouths among sharks. Green squares represents the thresher shark Alopias, which plots as having a narrower mouth relative to total length because of its elongate caudal fin.

When applied to Dunkleosteus (Table 1), mouth width produces estimates comparable to those estimated via UJP, if slightly (∼20%) lower. This further suggests mouth width and UJP are capturing the same morphometric signal. The slightly higher estimate for UJP may be because mouth perimeter is proportional to the two semi-major axes of an ellipse (Ramanujan, 1962), and if arthrodires have larger mouths than sharks in both anteroposterior and mediolateral axes (see below), then UJP will produce slightly larger estimates of body length than mouth width. When applied to Coccosteus cuspidatus, mouth width produces an estimated total length of 89.1 cm. This length is roughly twice the actual length of the animal, but similar to (if slightly lower than) the results using UJP, resembling the results for Dunkleosteus.

When log mouth width is graphed against log total length (Fig. 3), Millerosteus, Coccosteus, Incisoscutum, Watsonosteus, Plourdosteus, and Holonema plot far outside the 95% prediction interval for extant elasmobranchs, with mouths much wider than expected for their size. Mouth width generally underpredicts body length in complete arthrodires by 50% (File S4: table 6.1). The fact that arthrodires consistently have larger mouths relative to sharks and do so to a similar degree suggests it is unlikely this result could be attributed to the use of composite specimens or other sources of error. In order for an arthrodire like C. cuspidatus to have mouth proportions similar to modern sharks this taxon would require a mouth only 2.96 cm in width, about half its actual value (File S4: table 6.2). This occurs regardless of whether mouth width is measured from the internal or external angles (File S4: tab. 6.1). The only extant chondrichthyans that have mouth proportions comparable to arthrodires are Cephaloscyllium, Planonasus, and Isistius, which have unusually wide mouths for sharks (mouth width 13–15% total length, as opposed to 6–11% in most sharks) and thus do not approximate the normal relationship between body size and mouth width in Chondrichthyes. Based on the limited evidence and phylogenetic distribution of the taxa considered, it seems the idea that arthrodires had larger mouths than sharks should be the null hypothesis for future analyses.

Although most of these data are drawn from coccosteomorph arthrodires, data from Amazichthys trinajsticae (Jobbins et al., 2022) and an unpublished complete aspinothoracid in the collections of the CMNH (CMNH 50233) suggests this phenomenon is consistent across arthrodires. These specimens are either missing or have unprepared mouthparts, but the dimensions of their skull roof (assuming the cheek plates were located in a similar position as other arthrodires) suggest they have mouth proportions more similar to coccosteomorphs than extant sharks (at or outside the 95% PI for sharks, see the magenta stars in Fig. 3). These animals might be expected to have slightly wider mouths if the suborbital plates were preserved in situ, positioning these taxa even closer to other arthrodires. Thus, Dunkleosteus would be expected to show the same pattern.

Adding Dunkleosteus terrelli to Fig. 3 using the estimated lengths based on UJP results in this taxon plotting well within the prediction interval of extant sharks, unlike all other arthrodires examined. Indeed, length estimates based on UJP imply Dunkleosteus has a smaller-than-average mouth compared to extant sharks. However, given these lengths were calculated assuming Dunkleosteus had mouth-to-body proportions similar to extant sharks, such results are to be expected. Using internal mouth width results in Dunkleosteus having a disproportionately small mouth relative to extant sharks (see File S4).

UJP produces much narrower prediction intervals than mouth width (Table 1), but this is a consequence of how the models were fitted. UJP was fitted using a non-log transformed equation, following Ferrón, Martinez-Perez & Botella (2017) and Lowry et al. (2009), and was based on a narrower size range of large sharks. By contrast, the model for mouth width calculated here was based on a much wider body size distribution of shark taxa. Because of this the data for mouth width had to be log-transformed due to heteroskedasticity (i.e., increasing imprecision and error in data at larger sizes, resulting in bias in the best-fit line; see Kaufman, 2013), and indeed a non-transformed model showed significant heteroskedasticity of the residuals (File S4: fig. 8.2). The UJP model also showed signs of heteroskedasticity (File S4: fig. 8.1), though this was masked by the fact that there were few large sharks similar in size to Carcharodon carcharias. A good demonstration of the heteroskedasticity in the UJP regression model is that 95% prediction intervals are ±70% of the estimated length of Coccosteus cuspidatus yet only ±10.8% of the estimated length for CMNH 5768 (Table 1). This occurs despite the magnitude of the prediction intervals being the same for both: ±70 cm.

Predicting on log-transformed data nearly always results in much wider prediction intervals, often to the point of making the results uninformative, due to back-transforming the residuals to an arithmetic scale and changing their distribution from a normal distribution to a leptokurtic one. Calculations of prediction intervals are known to be very sensitive to kurtosis (Bonett, 2006; Miller, 1986), and hence log-transformed prediction intervals tend to be inflated. However, the smaller prediction intervals of UJP does not make it a more accurate predictor of size in Dunkleosteus. This is a question of accuracy over precision (National Academies of Sciences, Engineering, and Medicine, 2019: p. 47-51): it does not matter if a non-log transformed model is more precise if the model cannot accurately predict size in arthrodires in the first place.

Data for osteichthyans are not comprehensive, but bony fishes show much greater variation in relative mouth width than extant sharks (File S4: fig. 6.3). The present sample is not large enough to make broader conclusions about how relative mouth width varies across bony fishes, but at least some groups of bony fishes (e.g., Salmonidae, Esocidae, and Lepisosteidae) have mouth widths comparable to sharks, whereas others have mouths that are wider (e.g., Siluriformes, Centrarchidae, and Percidae) or narrower (e.g., Clupeidae) relative to their body size. This casts doubt on the idea that mouth dimensions reliably scale with body size across all fishes. For example, UJP is measurable in the large pimelodid catfish Brachyplatystoma capapretum (see Lundberg & Akama, 2005: fig. 8). The holotype specimen, MZUSP 78481, is a fish of ∼73 cm total length (58.1 cm precaudal length and 65.4 cm fork length, but has a damaged caudal fin). The UJP of this specimen is 13.7 cm, which produces an estimated total length of 121 cm using the UJP model, roughly 66% longer than the actual animal (File S4: table 5.3). This highlights that some fishes simply have proportionally larger mouths than extant sharks, and therefore mouth dimensions may not be universally scalable across fishes.

Notably, while arthrodires have mouths that are relatively wider than sharks, they fall within the range of variation of some groups of bony fishes, specifically catfishes (Siluriformes), especially predatory catfish clades (e.g., Pimelodidae, Bagridae, or Ictaluridae). Like arthrodires, these fishes also have relatively large mouths for their body size (File S4: fig. 6.3), though their mouths are anteroposteriorly narrow. Therefore, although arthrodires have larger mouths than sharks, their body plans do not require invoking proportions significantly outside the range of variation seen in extant fishes. Arthrodires do have disproportionately wide mouths among fishes in general, but their proportions do resemble at least one group of living fishes: catfishes.

Estimated UJP

Estimated UJP correlates strongly with total length in sharks (r2 = 0.86) and on average predicts total length within ±21.3% of the actual value (File S4: section 7). The scale-location plot of a model with an untransformed correlation between these variables shows significant heteroskedasticity (File S4: fig. 7.1). This demonstrates the larger span of body sizes considered here requires the data be log-transformed before analysis, even though the UJP equation of Lowry et al. (2009) did not show strong heteroskedasticity. Estimated UJP closely approximates actual UJP in the FSBC sharks (∼5% of actual value). The residuals versus fitted plot shows a large amount of variation and lacks a clear linear pattern (File S4: fig. 7.2). Much of this is driven by easily identifiable outliers, such as Alopias, Megachasma, and taxa with anteroposteriorly short mouths (Orectolobiformes, Squaliformes). Removing these taxa produces a better fit, though there are still outlying taxa whose removal cannot be easily justified (Hemigaleidae, Triakidae, Hexanchiidae, Cephalloscyllium, Odontaspis). Estimated UJP also did not accurately predict body length in Cladoselache, which appears to have a much larger UJP relative to body size than extant sharks (File S4: tab. 7.4) This, again, suggests there is not a universal correlation between mouth perimeter and body size across fishes.

The model using approximated UJP and all taxa (Table 2) produced lengths slightly less than the estimates of Ferrón, Martinez-Perez & Botella (2017), though estimated lengths for Coccosteus under this model are still more than twice the actual value. The model estimating length using approximated UJP and only considering predatory members of Carcharhiniformes, Lamniformes, and Hexanchiiformes (Table 2) almost perfectly replicated length estimates for Dunkleosteus in Ferrón, Martinez-Perez & Botella (2017), predicted values being within 5% of that study. However, values for Coccosteus are significantly lower than those using the UJP equation of Lowry et al. (2009), though they resemble the predicted values using mouth width. This is to be expected, as applying the model of Lowry et al. (2009) to Coccosteus requires some degree of extrapolation, whereas the model using approximated UJP contains more sharks in the size range of this taxon. However, the estimated length in the model only considering Carcharhiniformes, Lamniformes, and Hexanchiiformes was still over twice the measured value.

Table 2 Total length estimates for arthrodires using estimated UJP of extant sharks.

Total length calculated in these specimens using actual UJP of arthrodires and UJP estimated using Ramanujan’s approximation in a sample of extant sharks. % Dif represents the percent difference between these estimates and the UJP-based estimates of Ferrón, Martinez-Perez & Botella (2017). All measurements in cm.

Specimen	Taxon	Using all sharks	Macropredatory sharks only	
		Est.	% Dif.	Est.	% Dif.	
N/A	Coccosteus cuspidatus	83.1 (49.6–139.3)	−21%	79.8 (54.0–117.7)	−26%	
CMNH 7424	Dunkleosteus terrelli	302.1 (179.9–507.4)	−6%	318.5 (215.3–471.3)	−1%	
CMNH 6090	Dunkleosteus terrelli	475.0 (282.3–799.1)	−12%	517.6 (349.1–767.5)	−3%	
CMNH 7054	Dunkleosteus terrelli	507.3 (301.5–853.8)	−14%	555.5 (374.5–824.0)	−4%	
CMNH 5768	Dunkleosteus terrelli	589.6 (350.1–993.0)	−17%	652.7 (439.6–969.1)	−5%	

Graphing estimated UJP against total length produces similar results to mouth width: arthrodires have much larger mouths than sharks and are more similar to osteichthyans (File S4: fig. 7.4), though unlike Fig. 3 arthrodires do not plot near the extremes of variation within Osteichthyes. This may be because arthrodires tend to have short snouts (Engelman, 2023), and the elongate rostra of many osteichthyans (as well as the compressiform body shape of many species) results in them having seemingly larger mouths when UJP is scaled by total length. For example, Siluriformes, which stand out among fishes in having unusually wide mouths, are unremarkable among osteichthyans in estimated UJP because their mouths are anteroposteriorly short. However, the more elongate rostra of many osteichthyans are unlikely to correspond with an ability to swallow larger prey the way mouth width does (Mihalitsis & Bellwood, 2017). The ratio of estimated UJP to total length in Coccosteus (0.248, File S4: tab. 7.5) is comparable to Micropterus dolomieu (0.250). Coccosteus also shows a mouth size similar to salmonids (Salvelinus, Oncorhynchus), esocids (Esox), and some percids (Sander), but as these taxa are characterized by elongate snouts the utility of this comparison is unclear.

Head length

The length estimates produced by UJP and mouth width also require invoking highly unusual body proportions for Dunkleosteus. Specifically, reconstructing Dunkleosteus using length estimates from UJP and the caudal fin shape in Ferrón, Martinez-Perez & Botella (2017) results in an animal with an extremely short head relative to its body length and a hyper-elongate trunk region (Fig. 4). These proportions seem especially unlikely given Ferrón, Martinez-Perez & Botella (2017) also suggested Dunkleosteus likely had a short, lamnid-like body plan with a strongly lunate caudal fin based on ecological comparisons with extant sharks. Assuming the body length estimates based on mouth dimensions are accurate, this would imply Dunkleosteus terrelli had a head only 8% of its total length. These head-body proportions are seen in few other fishes, living or extinct. Modern fishes vary significantly in their head-body proportions, but most species have a head that ranges between 18–30% of total length (Fig. 5). Most taxa with heads less than 17% total length have highly elongate or anguilliform body plans with long trunks (Fig. 5), or else have highly elongate caudal fins (e.g., chimaeroids, Alopias spp., or Stegostoma) which dramatically increase body length relative to head length.

Figure 4 Reconstructed proportions of specimens of Dunkleosteus terrelli using the total lengths estimated by Ferrón, Martinez-Perez & Botella (2017) using UJP.

(A), CMNH 5768; (B), CMNH 6090.

Figure 5 Plot of percent head length against total length in fishes.

This graph shows how the estimates of Ferrón, Martinez-Perez & Botella (2017) result in unusually small head proportions for Dunkleosteus compared to other fishes. Taxa below the dashed line typically exhibit highly elongate trunks (e.g., anguilliform taxa, Cheirocentrus spp., Coryphaena spp.), or else have extremely elongate caudal fins that contribute greatly to total length (e.g., Alopias, Chimaeriformes). Regalecus silhouette modified from image by John Norris Wood, Coryphaena silhouette modified from image by Richard Winterbottom, and Ruvettus and Epinephelus silhouettes modified from images in Randall (1997) (CC-BY-NC 3.0: https://fishbase.mnhn.fr/photos/PicturesSummary.php?StartRow=0&ID=10411&what=species&TotRec=2, https://www.fishbase.se/photos/PicturesSummary.php?StartRow=1&ID=1044&what=species&TotRec=6).

However, the estimated lengths of Dunkleosteus using mouth dimensions require an even smaller head relative to its body than most eel-like fishes, in which head length is typically 12% or more of total length (Fig. 5). The only fishes which have head-body proportions comparable to those required for previous length estimates of Dunkleosteus are electric eels (Electrophorus spp.; head length 8.9–11.2% total length), and oarfishes (Regalecus spp.; 4.7–5.9% total length), both of which exhibit extreme head-body proportions even relative to other anguilliform fishes. The fishes which most closely resemble predicted head-body proportions for Dunkleosteus yet are not anguilliform or macruriform are taxa known for extremely elongate trunk proportions, including wolf-herrings (Cheirocentrus spp.; head length 13.3–14.2% of total length) and dolphinfishes (Coryphaena spp.; head length 15.6–16.4% of total length). However, the anatomy of Dunkleosteus shows no features that might indicate a highly elongate trunk, such as the tubular body armor of some camuropiscid coccosteomorphs (Dennis & Miles, 1979), or the elongate ventral shield of in some aspinothoracids (Jobbins et al., 2022). Not even Xiphactinus, which is a large-bodied fish characterized by a short head and an elongate trunk region, has proportions comparable to what the estimates using UJP imply for Dunkleosteus. The head of Xiphactinus is about ∼16.7% of total length, similar to other fishes with elongate trunks.

Notably, the unusual head-body proportions for Dunkleosteus in these reconstructions cannot be attributed to anatomical differences between arthrodires and other fishes. Arthrodires for which complete remains are known show head-body proportions well within the range of variation of modern fishes (Fig. 5), albeit on the lower end of the spectrum (∼20% total length). The only taxa that show shorter heads are Holonema (head length 15.9% total length) and Amazichthys (head length 15.1% total length), the latter of which plots among taxa with highly elongate trunks like Spanish mackerels (Scomberomorus spp.; head length 17.7–19.5% total length) and dolphinfishes (Coryphaena spp.). However, Amazichthys and Holonema are still well within the morphospace of taxa with elongate trunks, and are relatively close to the morphospace occupied by fusiform fishes. Thus, Dunkleosteus would be expected to conform to the head-body proportions seen in other arthrodires and fusiform fishes more broadly, as this taxon shows few features indicative of an autapomorphic, hyper-anguiliform body plan.

Predicting total length using head length results in an estimated length of only 2.69 m for CMNH 5768. Although there is some reason to believe head length by itself may not be ideal for estimating body length in arthrodires (mostly stemming from their shorter snout), these estimates are so divergent from those calculated using mouth dimensions (2.69 m using head length; 6.88 m using UJP) it suggests one or both of these estimates is likely to be wrong. Given head length more closely correlates with total length across fishes (including arthrodires) and arthrodires have much larger mouths than expected for their body size, this suggests body size estimates using mouth dimensions (UJP, mouth width) are overestimates and the 2.69 m estimate via head length is closer to the actual value. More extensive analyses of length in Dunkleosteus are currently under preparation by the present author (Engelman, 2022a; Engelman, 2023) and discussing them is beyond the scope of this mouth dimension-focused study.

Discussion

Estimating length in Dunkleosteus terrelli

The size of Dunkleosteus terrelli has historically not been estimated with statistical rigor (see discussion in Ferrón, Martinez-Perez & Botella, 2017). Most previous size estimates of Dunkleosteus are speculative “guesstimates” that simply state the size of the animal without clarifying how this estimate was calculated or what prior study this value was taken from (e.g., Anderson & Westneat, 2007; Anderson & Westneat, 2009; Carr, 2010; Denison, 1978; Long, 2010; Newberry, 1873). In most of these cases, the unstated assumption is these values were calculated based on the dimensions of smaller arthrodires like Coccosteus cuspidatus, but even in these cases it is usually not reported what anatomical measurements were used to estimate body size. One of the goals of Ferrón, Martinez-Perez & Botella (2017) was to rectify this issue, by producing estimates of body size in Dunkleosteus based on rigorous statistical criteria. However, when the length estimates of Ferrón, Martinez-Perez & Botella (2017) are used to create reconstructions of Dunkleosteus, they result in proportions so dissimilar to any fish, living or extinct (including other arthrodires), they appear biologically implausible (Figs. 4–5). The resulting hyper-anguilliform body plan produced by these methods is especially unusual as Ferrón, Martinez-Perez & Botella (2017) also predicted Dunkleosteus to show an anteroposteriorly short body plan with a lunate caudal fin, similar to modern lamnids. Hence, the results of Ferrón, Martinez-Perez & Botella (2017) would theoretically favor a much shorter length and higher fineness ratio for the body of Dunkleosteus than previously predicted. Indeed, if scaling off of the reconstruction in Ferrón, Martinez-Perez & Botella (2017: Fig. 4B) using the known head length of CMNH 5768 (61.3 cm; measured as the greatest head length from the cranio-thoracic joint to the tip of the rostral in lateral view) produces a length of only 3.77 m, supporting shorter lengths for D. terrelli. It should also be noted Ferrón, Martinez-Perez & Botella (2017) did not intend their length estimates to be taken as an exhaustive examination of body size in D. terrelli (H.G. Ferrón, pers. comm., 2022). Instead, the primary concern of these authors was to provide a ballpark size estimate for this taxon to allow them to investigate their real question of interest, caudal fin shape, as total length was a necessary input to control for potential allometric shape variation (H.G. Ferrón, pers. comm., 2022; Ferrón, Martinez-Perez & Botella, 2017: p. 5).

The reconstructed body shape of Dunkleosteus using previously cited lengths (Fig. 4) would also impose severe biomechanical and physiological penalties on the animal. In this reconstruction, the deepest part of the body is located at ∼20% total length, whereas in most aquatic vertebrates this value is ∼36% total length and is typically close to the anteroposterior level of the center of mass (Dean, 1902; Parsons, 1888). This is thought to be a constraint related to maintaining a hydrodynamic body shape. Similarly, the gill chamber in this reconstruction of Dunkleosteus is very small for its body size (= body volume). This would make it difficult for the animal to obtain enough oxygen to support its metabolic needs (Alexander, 1967; Schmidt-Nielsen, 1984). Some elongate-bodied fishes do have smaller gill chambers, but these species also tend to have dorsoventrally shallow or mediolaterally narrow bodies, which reduces the amount of biomass relative to their length. Arthrodires, by contrast, have thoracic armors which suggest a very deep and wide body relative to the head (Engelman, 2023).

Most interpretations of body shape and body proportions in arthrodires rely, by necessity, on Coccosteus cuspidatus, one of the few arthrodires known from near-complete remains (Miles & Westoll, 1968). However, using Coccosteus as a Rosetta Stone to extrapolate the proportions of other arthrodires can be fraught with peril, as discussed by Ferrón, Martinez-Perez & Botella (2017). These authors rightfully pointed out the over-reliance on Coccosteus cuspidatus in previous studies when reconstructing the appearance and proportions of larger arthrodires. Specifically, given Coccosteus is a freshwater/brackish demersal taxon (roughly analogous to Triaenodon in habits among living sharks) the proportions of this species would not be expected to be indicative of large, pelagic placoderms like Dunkleosteus. This is especially the case given the body shape of sharks shows a high degree of ecological lability (Ferrón, Martinez-Perez & Botella, 2017; Jobbins et al., 2022). Variability in the body shape of arthrodires can even be seen when comparing postcranial proportions among the limited overlapping material between Coccosteus cuspidatus and other taxa. In Coccosteus the thoracic shield is slightly wider than tall, with a flat ventral bottom (Miles & Westoll, 1968), as expected of a demersal taxon that spent significant amounts of time resting on the substrate. By contrast, many Gogo arthrodires (Dennis & Miles, 1979; Miles & Dennis, 1979: fig. 8; Trinajstic, 1995) and Cleveland Shale taxa (Heintz, 1932; D Chapman pers. comm., 2014; Engelman pers. obs., 2022), which are considered to be more active swimmers (Carr, 2010; Trinajstic, Briggs & Long, 2022a), have a thorax that is subcircular in cross-section and a ventral armor that is often strongly curved (Fig. 6).

Figure 6 Thoracic armors of arthrodires in anterior view.

(A) Coccosteus cuspidatus (from Miles & Westoll, 1968: fig. 44; ©Cambridge University Press) and (B) Harrytoombsia elegans (from Miles & Dennis, 1979: fig. 9; ©Oxford University Press), showing the flat ventral armor of the demersal, freshwater/brackish water C. cuspidatus compared to the curved ventral armor of the marine, reef-dwelling H. elegans. Scale = 3 cm.

The present study also relies heavily on comparisons between Dunkleosteus and Coccosteus or other, similar arthrodires, and thus might be criticized for falling into the same fallacy that the anatomy of Coccosteus is a reliable analogy for Dunkleosteus. However, the discrepancies between Coccosteus and sharks do not appear to be driven by changes related to ecology and body shape, but rather by features that differentiate arthrodires from elasmobranchs (specifically, mouth width and head shape). Pelagic arthrodires (Dunkleosteus) and pelagic sharks both show mouth proportions more similar to their demersal relatives than each other (and vice versa). Additionally, the head-body proportions of sharks and arthrodires agree with one another and disagree with previous estimated lengths for Dunkleosteus, regardless of life habits. Thus, although it could be argued Dunkleosteus exhibited different head-body proportions from Coccosteus, current length estimates require invoking proportions for Dunkleosteus that are not only unsupported by the known anatomy of D. terrelli but also aberrant for fishes and arthrodires as a whole.

Mouth size in arthrodires

One interesting conclusion of this study is arthrodires seem to be characterized by much larger mouths relative to their body size than sharks. Arthrodire paleobiology has generally been reconstructed based on comparisons with extant chondrichthyans (e.g., Carr, 2010; Ferrón, Martinez-Perez & Botella, 2017; Long, Trinajstic & Johanson, 2009; Trinajstic et al., 2022b), largely due to inferred similarities in functional anatomy (Carr, Lelièvre & Jackson, 2010), paleoecology (Carr, 2010), and life history (Long, Trinajstic & Johanson, 2009). However, the present study indicates one way arthrodires are unlike sharks: specifically in having much larger mouths relative to body size. Thus, using mouth proportions of extant sharks to estimate body size in arthrodires produces significant overestimates.

These difficulties in estimating the body size of arthrodires are similar to historical issues in mammalian paleontology estimating the body size of extinct carnivorous mammals such as Hyaenodonta. Like arthrodires, hyaenodonts are an entirely extinct group of predatory vertebrates whose paleobiology, by necessity, has been reconstructed based on their closest modern ecological analogues: the Carnivora in the latter’s case (Egi, 2001; Matthew, 1909; Morlo, 1999; Scott & Jepsen, 1936; Van Valkenburgh, 1988). This has been done despite several studies noting hyaenodonts often exhibit morphological features unlike any living carnivoran (Borths & Stevens, 2017; De Iuliis, 1993; Mellett, 1969; Mellett, 1977). One way in which inferring the paleobiology of hyaenodonts based on distantly related, anatomically distinct animals can be problematic is when complete skeletons of hyaenodonts are discovered they invariably show disproportionately large heads relative to extant mammals (Egi, 2001; Engelman, 2022b; Scott & Jepsen, 1936; Van Valkenburgh, 1990). As a result, body size estimates in hyaenodonts based on cranio-dental variables invariably result in major overestimates of body mass (Savage, 1977; Van Valkenburgh, 1987; Van Valkenburgh, 1990).

It is rather unusual that in both of these cases the earlier-diversifying clade is characterized by disproportionately large heads or mouths (arthrodires and hyaenodonts), whereas their later ecological analogues (neoselachian sharks and carnivorans) have smaller mouths and heads. This phenomenon may not be restricted to fishes and mammals: erythrosuchids, which represent some of the earliest archosaurian experiments in macrocarnivory, also have disproportionately large heads compared to later predatory archosaurs such as dinosaurs and “rauisuchians” (Bestwick et al., 2022). Large heads and mouths characterize a number of (mostly extinct) vertebrate clades, but their functional or evolutionary significance is unclear (Bestwick et al., 2022). It is possible these traits are a side-effect of early burst diversification. When clades experience an adaptive radiation one of the simplest and most evolutionarily rapid ways to produce a macropredator morphotype might be for a generalist taxon to develop a hypertrophied mouth and/or head relative to body size. This allows the animal to take much larger prey relative to its body size without significantly altering its functional morphology, developing novel or complex adaptations, or engaging in the costs of large body size. Whether this represents a broader phenomenon among vertebrates is intriguing but beyond the scope of the present study. However, older clades having larger heads may not be a universal pattern. Groupers (Serranidae), one of the most ecologically successful groups of extant fishes, also have unusually large heads relative to body size (File S4: fig. 9.5).

The disproportionately large mouths of arthrodires are further demonstrated by the bony anatomy of these taxa. In most fishes, including sharks, the angles of the mouth are located at or anterior to the posterior edge of the neurocranium, and the mouth does not encompass the entire anteroposterior length of the head (Figs. 2A–2B). This also results reduces mouth width, as the head is typically narrower anteriorly and the angles of the jaw cannot be placed more laterally. On the other hand arthrodires, especially pachyosteomorph arthrodires, have a mandibular joint located near the very posterior end of the skull, with the inferognathal and cranium articulating on the postsuborbital plate of the cheek (Carr, 1991; Heintz, 1932). This results in the mouth extending most of the anteroposterior length of the skull (Figs. 2C–2D) and thus arthrodires have very large gapes relative to their size. An analogous pattern can be seen in several groups of extant fishes, including dragonfishes and viperfishes (Stomiidae), lizardfishes (Aulopifomes), and dogtooth characins like payaras (Hydrolycidae), in which the jaw articulation is shifted posteroventrally relative to the dorsal boundary of the suborbital/postsuborbital and the opercular series, the preopercle obliquely oriented to the anteroposterior axis, and the “cheek” (roughly equivalent to the cheek unit in arthrodires) has a sharp posteroventral angle (Fig. 7). This results in a large gape relative to head size (Fig. 8). This morphology may be associated with a tendency to take comparatively large prey relative to the predator’s body size, at least based on observations of extant stomiids (Clarke, 1982; Kenaley, 2012) and hydrolycids (Goulding, 1980). Thus, even though length estimates for Dunkleosteus are not provided here, this animal likely resembled other arthrodires in having a disproportionately huge mouth relative to body size.

Figure 7 Examples of living fishes with a posteroventrally located jaw joint.

(A) viperfish, (Chauliodus macouni, modified from Kenaley, 2012), and (B) lizardfish (Saurida wanieso, redrawn from Reyes (2007), CC-BY-NC 3.0: https://fishbase.mnhn.fr/photos/PicturesSummary.php?StartRow=0&ID=10411&what=species&TotRec=2), showing how the more posterior jaw articulation in these species results in a larger mouth size relative to head length, compare to Fig. 2.

Figure 8 Comparative gape size in fishes.

Heads of (A) viperfish, (Chauliodus macouni, modified from Kenaley, 2012), (B) Dunkleosteus terrelli (modeled after CMNH 6090), and (C) shark (Carcharhinus obscurus, proportions from Garrick, 1982), showing how the more posterior articulation of the jaw in A–B allows a proportionally wider gape (denoted by green arrow) relative to head/organismal size. Chauliodus has a wider gape than Dunkleosteus, but the tall, needle-like teeth result in similar clearances between the jaws. Gape angle in Dunkleosteus is conservative and based on the degree the jaws could be opened before the gape angle interfered with function of the adductor mandibulae (using muscle reconstructions from Long, 1995).

In fact, Dunkleosteus terrelli appears to have a large mouth even compared to other arthrodires. Comparing anteroposterior mouth length as a percentage of head length finds Dunkleosteus has a much larger mouth than most extant fishes (File S4: fig. 9.1). Dunkleosteus also has a larger mouth than any other arthrodire examined, including Millerosteus minor, Coccosteus cuspidatus, Plourdosteus canadensis, the Gogo Formation coccosteomorphs, and the late Devonian (Frasnian-Famennian) Bungartius perissus, Heintzicithys gouldii, Draconichthys elegans, Gymnotrachelus hydei, Hadrosteus rapax, and Pholidosteus friedeli. Dunkleosteus also has a much larger mouth than the Gogo dunkleosteoid Eastmanosteus calliaspis, suggesting these patterns are not driven by differences in head shape (specifically, differences in the location of the cranio-thoracic joint between coccosteomorphs and dunkleosteoids). This also suggests estimating the length of Dunkleosteus using the gnathal dimensions of coccosteomorphs (as some previous studies have implied) is likely to produce overestimates.

Omitting narrow-snouted taxa like needlefishes (Belonidae) and gars (Lepisosteidae), adult Dunkleosteus have longer mouths relative to their head than many extant fishes such as groupers, though not quite as large as the mouths seen in dragonfishes (Stomiiformes), sabertooth anchovies (Lycengraulis spp.), and lizardfish (Synodontidae). Dunkleosteus shows a positive allometry in mouth length, with the largest specimens (CMNH 5768 and CMNH 7054) having much larger mouths than the smallest ones (CMNH 7424 and CMNH 6090). This is a common trend in fishes with large gapes, including flathead catfish (Pylodictis olivaris; Slaughter & Jacobson, 2008), black basses (Micropterus spp.; Moran, Ward & Gibb, 2018), and is somewhat supported by the data for sharks in the present study.

The disproportionately wide mouths of eubrachythoracid arthrodires have further implications for their paleobiology. In living fishes, one of the biggest determining factors of prey choice is mouth size. This constraint is most clearly seen in extant gape-limited actinopterygians, but also applies to “dismemberment” predators with cutting mouthparts such as sharks (including squaloids; Bangley & Rulifson, 2011), piranhas, barracudas, and by analogy eubrachythoracid arthrodires with bladed mouthparts like Dunkleosteus. “Dismemberment” predators typically rip chunks out of their prey when they feed or break prey into manageable pieces, and hence are able to consume animals larger than the diameter of their mouth (Grubich, Rice & Westneat, 2008; Helfman & Clark, 1986). However, there is still an indirect constraint present in that mouth size affects an animal’s ability to capture, handle, and process prey (Lucifora et al., 2008). A larger mouth relative to body size means a fish can swallow bigger prey items, apply more bite force to restrain or incapacitate prey, take larger bites out of prey producing faster blood loss or shock, and overall feed on larger animals relative to its body size. Given arthrodires in this study are found to have much larger mouths than sharks of a similar size, these animals may have been able to take larger prey relative to their body size than extant sharks.

To describe the effect large mouths have on the paleoecology of arthrodires in more intuitive terms, an individual of Dunkleosteus the size of CMNH 5768 could potentially consume a bolus of food 20,803 cm3 in volume (length of biting edge of inferognathal = 23.9 cm, width between gnathal articulations = 53.8 cm, approximate clearance between caniniform fangs of the anterior supragnathal and inferognathal when mouth is reconstructed in an open position = 30.9 cm, modeling the resulting value as an ellipsoid). Converting this value to mass by multiplying by the density of muscle (1.1 g/cm3), this means that an adult individual of D. terrelli would be able to tear off and swallow roughly 22.9 kg of biomass in a single bite. This might be a slight overestimate given mouth width would be narrower in life due to soft tissues, but even reducing mouth width by 5 cm only reduces estimated bite volume by 5%. This value is 37% the average body mass of a human being (Walpole et al., 2012) and is comparable to rough estimates of bite volume (∼23500 cm3/25.9 kg) for the largest reliably measured individual of C. carcharias (MZL 23981; De Maddalena, Glaizot & Olivier, 2003). MZL 23981 has a total length of 5.65 m and an estimated weight of nearly 2 tons (De Maddalena, Glaizot & Olivier, 2003; Mollet & Cailliet, 1996), and has a mouth width (50 cm) and length (30 cm) comparable to CMNH 5768 (variation in these estimates is mostly due to uncertainty in reconstructed gape angle and whether the gnathal fangs impede gape). The size of Dunkleosteus is still difficult to determine but the present analysis and other studies currently in preparation (Engelman, 2022a; Engelman, 2023) suggest CMNH 5768 was substantially smaller than MZL 23981. Thus, CMNH 5768 could have bitten off chunks of flesh comparable to those produced by the largest known great white sharks despite being substantially smaller in size.

These inferences are supported by an arthrodire specimen referred to Holdenius holdeni (CMNH 8130) that apparently died while attempting to prey upon a large ctenacanth shark (Hlavin, 1990). The ctenacanth was reported as 2 m in estimated length based on the size of the spine fragments whereas the predating arthrodire was estimated to be approximately 3 m in length (Hlavin, 1990). Examination of the specimen by the present author suggests the preliminary length estimate is reasonable for the arthrodire, but the author is not familiar enough with ctenacanth spine proportions to speak on this estimate. The arthrodire is rather small relative to its would be-prey. Although this is a case of failed predation, most fishes would not even attempt to attack an animal 2/3 their own body length, some rare and exceptional behavior in taxa like Carcharodon carcharias notwithstanding (Dines & Gennari, 2020). This is not the only instance of large arthrodires attacking large fishes, bite marks from large arthrodires have also been reported on specimens of large dunkleosteids (Capasso et al., 1996; Hall, Ryan & Scott, 2016; Hussakof, 1906) and the large arthrodire Titanichthys (Anderson & Westneat, 2009: p. 266). At least some of these bite marks (i.e., those on Titanichthys) do not seem explainable by intraspecific combat. This further implies arthrodires may have functioned very differently from modern sharks in their paleoecology, with arthrodires feeding on comparatively larger fishes relative to their body length. Similar arguments may also apply to other groups of Devonian fishes, including chondrichthyans (i.e., ctenacanths, symmoriforms; Harris, 1938; Hodnett et al., 2021; see also File S3: tab. 7.4 of this study), actinopterygians (i.e., “palaeoniscoids”; Choo, 2015; Dunkle, 1964; Pearson & Westoll, 1979), and some groups of sarcopterygians (i.e., onychodontids; Long, 1991). Like arthrodires, these fishes also tend to have more posteriorly placed jaw joints and/or large mouths compared to their modern relatives, though not to the same extreme as arthrodires.

Among living fishes, large active-swimming catfishes such as pimelodids (e.g., Brachyplatystoma, Phractocephalus), might be better functional analogues for arthrodires than sharks, at least in some respects. Indeed, the anatomies and evolutionary histories of eubrachythoracid arthrodires and pimelodid catfishes are eerily similar in a number of respects. Both groups have much wider mouths relative to their body length than sharks (see present study). Both groups combine an extensive dermal skeleton which covers the first third to half of the body with an unarmored, scaleless post-thoracic region (Burrow & Turner, 1999). Both are largely demersal to nektonic predators that secondarily adapted to the lifestyle from benthic, detritivorous ancestors (Lebedev et al., 2009). Both show adaptations to suction feeding (Anderson & Westneat, 2007; Anderson & Westneat, 2009), unlike most large sharks (but not smaller taxa like squaloids; see Wilga & Ferry, 2015). Some of the few noteworthy differences between the two groups from an ecological/functional morphological perspective are that pimelodids have much smaller eyes, are gape-limited predators with patches of flattened villiform teeth used for gripping prey (rather than bony fangs and blades that allow for oral processing as in arthrodires), and pimelodids are exclusively freshwater whereas eubrachythoracid arthrodires primarily inhabit marine or brackish waters (with many localities traditionally considered as freshwater having been reinterpreted as brackish or coastal marine environments; Hamilton & Trewin, 1988; Schultze & Cloutier, 1996; Wilson et al., 2014). Therefore, at least in some respects, large nektonic catfishes such as pimelodids may prove to be better paleoecological analogues for arthrodires than sharks.

In conclusion, length estimates for Dunkleosteus terrelli using mouth dimensions appear unlikely for several reasons. First, when applied to smaller arthrodires for which complete remains are known they produce inaccurate results. Second, these methods result in reconstructions of Dunkleosteus that have unusually short heads and hyper-anguilliform proportions relative to other fishes, whereas complete arthrodires show proportions similar to other fusiform fishes. Third, the assumption that sharks and arthrodires show similar mouth-body proportions does not hold; arthrodires have much larger mouths than sharks of similar body length, more comparable to nektonic catfishes. Together, these observations suggest that length estimates commonly cited for Dunkleosteus are overestimates, and this taxon may not have reached lengths of 8 m. Additionally, despite the criticism of the length estimates in Ferrón, Martinez-Perez & Botella (2017) here, the broader results and conclusions of that study focusing on caudal fin shape in Dunkleosteus are likely correct. Indeed, evidence from other arthrodires (Jobbins et al., 2022) and other regions of the body in Dunkleosteus Engelman (2023) seems to support Ferrón, Martinez-Perez & Botella (2017)’s conclusions regarding caudal fin shape in this taxon. Determining exactly how large Dunkleosteus terrelli and other such arthrodires were is beyond the scope of the present study, but is currently under investigation by the present author (Engelman, 2022a; Engelman, 2023). Overall, this study highlights the uncertainties regarding body size estimation in placoderms, as well as the need for a more rigorous method of inferring body size in these taxa.

Supplemental Information

Supplemental Information 1 Dataset of morphometric data from extant fishes

Morphometric data used in the present study, including total length, head length, mouth width, and mouth length. References used in this file can be found in File S9.

Click here for additional data file.

Supplemental Information 2 Dataset of total length and upper jaw perimeter in sharks from Lowry et al. (2009)

Estimates of fork and precaudal length as described in the Materials and Method section.

Click here for additional data file.

Supplemental Information 3 Jaw perimeter data from sharks of known length measured at the Florida Fish and Wildlife Research Institute (FSBC)

Click here for additional data file.

Supplemental Information 4 Knitted .html document showing the results of all the analyses in this study

Original R code can be downloaded from this file as an .rmd document by selecting the option “Download Rmd” at the top of the document.

Click here for additional data file.

Supplemental Information 5 Silhouette of Epinephelus malabaricus

This image is used to produce Fig. 5, and necessary to rerun the code. Silhouette modified from image in Randall (1997).

Click here for additional data file.

Supplemental Information 6 Ordered bibliography of references used in File S1

Click here for additional data file.

Supplemental Information 7 .png silhouette of Regalecus

This image is used to produce Fig. 5, and necessary to rerun the code. Regalecus silhouette modified from image by John Norris Wood.

Click here for additional data file.

Supplemental Information 8 Silhouette of Coryphaena hippurus

This image is used to produce Fig. 5, and necessary to rerun the code. Corphaena silhouette modified from image by Richard Winterbottom (from https://www.fishbase.se/photos/PicturesSummary.php?StartRow=10&ID=6&what=species&TotRec=21, used with permission).

Click here for additional data file.

Supplemental Information 9 Silhouette of Ruvettus pretiosus

This image is used to produce Fig. 5, and necessary to rerun the code. Silhouette modified from image in Randall (1997).

Click here for additional data file.

Supplemental Information 10 Click here for additional data file.

I would like to thank the various curators and collections managers, including C Colleary, A McGee and H Majewski (CMNH, paleontology), R Muehlheim (CMNH, extant fishes), J Maisey and A Gishlick (AMNH), B Simpson, L Grande, and C McGarrity (FMNH), E Price (FSBC), M Daly and M Kibbey (OSU), J Kerr (MNHM), and S Walsh (NMS), for access to specimens in their care. I thank J Newman for providing photos of Millerosteus and Watsonosteus; C Engelman for assistance in taking measurements of preserved fishes; F Concha, R Freitas, F Mollen, and R Winterbottom for permission to use images of fishes they had collected; M Borths for discussions regarding hyaenodonts; D Chapman for discussions regarding the history of the CMNH; S Adnet, F Mollen, and P Jambura for discussions on mouth proportions in sharks; Y Haridy for assistance in getting measurements of FMNH specimens; and D Croft, S Simpson, M Benard, P Princehouse, N Vitek, R Drushel, N Gardner, R Shell, R Hawley, L Bernstein-Kurtycz, R Oldfield, J Lundberg, J Hannibal, M Mihalitsis, M Balk, O Lebedev, Y Haridy, Z Johanson, J Long, K Trinajstic, and M Jobbins for helpful feedback. Part of this study grew out of a failed study that attempted to apply the methods of Ferrón, Martinez-Perez & Botella (2017) to Titanichthys and other arthrodires, and I thank R Matsumoto and N Smith (my would be co-authors on this project) for discussions regarding mouth size in filter-feeding fishes that led to the present analysis. Finally, I thank HG Ferrón for providing clarification on the methods of their 2017 study.

Institutional abbreviations

AMNH American Museum of Natural History, New York, USA

CMNH Cleveland Museum of Natural History, Cleveland, OH, USA

FMNH Field Museum of Natural History, Chicago, Illinois, USA

FSBC Fish and Wildlife Research Institute, Florida Fish and Wildlife Conservation Commission, St. Petersburg, Florida, USA

LDUCZ Grant Museum of Zoology, University College, London, U.K

MNHM Musée d’Histoire Naturelle de Miguasha, Quebec, Canada

MZL Musée Cantonal de Zoologie, Lausanne, Switzerland

MZUSP Museu de Zoologia da Universidade de São Paulo, São Paulo, Brazil

NHMUK The Natural History Museum (London), London, U.K

NMS National Museum of Scotland, Edinburgh, UK

OSUM Ohio State University Museum of Biological Diversity, Columbus, Ohio, USA

ROM Royal Ontario Museum, Toronto, Ontario, Canada

WAM Western Australian Museum, Perth, Australia

Additional Information and Declarations

Competing Interests

Author Contributions

Data Availability

The authors declare there are no competing interests. There is no need to provide a field permit as all work was done on specimens in museum collections.

Russell Engelman conceived and designed the experiments, performed the experiments, analyzed the data, prepared figures and/or tables, authored or reviewed drafts of the article, and approved the final draft.

The following information was supplied regarding data availability:

The raw data are available in the Supplemental Files.

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
