# Peer review of "Giant, swimming mouths: oral dimensions of extant sharks do not accurately predict body size in Dunkleosteus terrelli (Placodermi: Arthrodira)"

_PeerJ, doi:10.7717/peerj.15131_

## Round 0.1 · original submission · Minor Revisions

Both reviewers are quite positive in their assessment of the manuscript, and have only relatively light edits to suggest. I am in agreement with their comments, and have nothing else to add. I look forward to reading the revised manuscript.

Reviewer 1 ·

Basic reporting

This is a really nice paper, providing needed context to previous reconstructions of Dunkleosteus terrelli. The metholodogical choices of Ferron et al. (2017) are thoroughly examined. It should provide a valuable resource for future studies on arthrodire ecology and biomechanics.

Clear and unambiguous, professional English used throughout.
- Yes – very few grammatical corrections to make
Literature references, sufficient field background/context provided.

- Yes – good summary of the literature, the relevance of the topic is well-explained. References have been checked.

Professional article structure, figures, tables. Raw data shared.
- Yes

Self-contained with relevant results to hypotheses.
- Yes

Experimental design

The experiment is well-structured. My only question would be regarding the use of mouth width as a proxy for UJP. The length estimates for placoderms based on the two values are described as 'relatively similar', but the differences for some are quite different (8 to 24%). Did the author consider using a value that incorporated both the mouth length and width to more closely approximate the UJP? Regardless, this doesn't have an impact on the primary conclusions, it just might be worth looking into if planning to apply scaling relationships based on mouth dimensions in future.

Original primary research within Aims and Scope of the journal.
- Yes

Research question well defined, relevant & meaningful. It is stated how research fills an identified knowledge gap.
- Yes

Rigorous investigation performed to a high technical & ethical standard.
- Yes

Methods described with sufficient detail & information to replicate.
- Yes. I really like the way all the code is laid out in a single R markdown file, with clear descriptions of every step and the analysis of the results. The code ran with no issues and produced the same results as the paper. The underlying data has all been made available and is clearly laid out.

Validity of the findings

Impact and novelty not assessed. Meaningful replication encouraged where rationale & benefit to literature is clearly stated.
- Yes

All underlying data have been provided; they are robust, statistically sound, & controlled.
- Yes

Conclusions are well stated, linked to original research question & limited to supporting results
- Yes

Additional comments

The paper is well-argued and provides an important addition to the literature, I recommend it for publication, with extremely minor revisions. Comments/corrections follow:

L64 – should be “the size of prey items eaten by a predator strongly correlates with both…”
L68 – the sentence should be rewritten into two separate sentences.
L77 – should be “vertebrates appear to display ranges of body size disparity comparable to modern marine life”
L79 – should be either “increases in atmospheric oxygen” or “increasing atmospheric oxygen”
L109 – needs a comma after “from this”
L195 – the sentence should begin “in Dunkleosteus”
L205-293 – fantastic detail which is important for replicability, but would it be better in a supplementary methods section instead of the main text?
L296 – not taxa, instead say specimens/individuals/etc.
L345 – the sentences about the .html document and the .rmd file might be better in a separate data availability section?
L449 – what is the rough mouth width:total length ratio for these genera?
L517 – these clades are not specifically labelled in supplementary file 4, fig 5.2 – please amend this
L558 – should be Figure 5, not Figure 4
In general, much of the Results section reads more like Discussion – although this isn’t necessarily a problem
L624 – capitalise “Figure”
L632 – Delete “That is”, just begin the sentence “With regards…”
L643 – Delete the first sentence, the point is repeated in L648
L936 – the reference should be amended to include the Hlavin section – it is included in the text
Fig 1 – B shows the right lateral view not the left lateral view
Fig 3 – the teal stars aren’t that distinct from the grey points, could a different colour be tried?
Fig 5 – it might be nice to have a couple of extra silhouettes next to the y axis to illustrate the differences in percent head length in extant fish
Fig 5 – the shapes in the legend could be made larger
Fig 6 – including the species names and scale bar dimensions in the figure itself might make it easier to quickly interpret
Fig 7 – in the legend (L6), “means” should be “mean that”
Fig 7 – what were the range of motion estimates based on? Are they, or the jaw muscle positions, taken from a published paper?
Fig 9 – is there any similarity between Dunkleosteus and the pimelodids? Or is that restricted to mouth width?
Table 1 – include units in the column headings
Table 4.1 in supplementary – for percentage difference column, report the differences as number not decimals (i.e. 13 rather than 0.13)

Reviewer 2 ·

Basic reporting

The manuscript reads well, it is written in professional and generally clear language, though at some points it needs rephrasing to improve comprehensibility and accuracy, see below:

Line 24: Please rephrase “and the Devonian more generally” which is a bit confusing at first read. Maybe consider adding another sentence;
Line 79-80: Please rephrase “increases atmospheric oxygen”. I suppose you meant “increase in atmospheric…” or “increased atmospheric…”;
Line 514: Please replace “mouth” with “mouths”;
Line 764-765: “The arthrodire is very large relative to its prey” Are you sure? Shouldn´t it be “the arthrodire is rather small…”?;
Line 872-885: Here Figures 7 and 8 are mixed up and a bit of text is missing in the legend of Fig. 8 (Lines 872-875).

The introduction and background are processed thoroughly and sufficiently, the numerous references are relevant. The methodology is precise, understandable, up-to-date, well explained and useful for future researchers. The figures and tables are clear, of good quality and most of them are relevant to the main conclusions of the study. As the diagram in Fig. 9 does not show any clear trend, you may consider including it among the supplementary data.

The study is based on an impressing amount of raw data, all of which are included in the submission. I couldn't check the R code because I cannot open .rmd files. I personally have no experience with statistical analyses in R, so I am not competent to comment on the details of this part of the methodology. If the other reviewer likewise, I suggest adding an additional reviewer with sufficient expertise.

Experimental design

The research question is well defined and useful, not only in terms of setting the limits of the primary inputs of morphometric analyses, but in a broader view outlining interpretations of the palaeoenvironments and the trophic roles of the organisms inhabiting them. It is really intriguing that the dimensions of this widely popular arthrodire were generally estimated only approximately, thus this valuable research definitely fills a gap in our knowledge. The methodology used in the manuscript is well justified, its constraints and advantages are thoroughly explained, model animals and data well chosen. Specific comments below:

Line 263: “suggesting bias from allometric scaling” – I don´t understand, please explain. Why is it suggested? Who suggests it?
Line 413-416 and elsewhere: Did you consider soft mouth parts (muscles, skin) when measuring UJP and mouth width in extant taxa? I see you mention that in case of fossils the presence of soft tissues should be considered (Line 749), but I didn´t find any info as to what exact morphometric points were chosen for measurements of extant specimens. I suggest to produce a drawing and add it to Fig. 1 or create a new figure.

Validity of the findings

It is fascinating, that in case of such a favorite (especially in the US) giant fossil predator, there has only been one attempt (by Ferron et al. 2017) to base its length estimates on sound analyses supported by data. The present manuscript may be considered as a response to Ferron et al., exhaustively explaining the limits and methodological shortcomings of their chosen analytical approach. And it is a very reasonable response, as the current phylogenetic consensus (e.g. Giles et al. 2015, Qiao et al. 2016) places arthrodires as stem-group osteichthyans, and thus more morphologically dissimilar to chondrichtyans than previously thought.
Especially, how come no one has ever attempted to try and draw the body outline of this supposed giant? Figure 4 of the present manuscript is a glowing example of how sometimes a simple method such as a shadow drawing can be so scientifically beneficial. An animal shaped like this could simply not exist and the present author is very convincing in their arguments. I really have to highlight this simplicity which stands out among all the hard-computerized data. It is a proof that the author really took their time to think comprehensively about the problem and I dare to say that with this approach they could deal with a really wide and diverse range of research gaps in the future. It is a bit of a shame that we don´t get to know at least an approximate estimate of the length of Dunkleosteus, instead we are referred to an upcoming publication. Perhaps it would be worth at least a percentage evaluation of how much the new estimate differs from the old ideas, the reader is curious.

The manuscript is quite long and individual aspects are discussed in detail and exhaustively. Nevertheless, most of the text has its purpose and it is not necessary to delete or edit it. Exception below:
Line 665-680: This part of the manuscript is speculative and, unlike the rest of the manuscript, unsupported by data. As it doesn´t upgrade the main results and it is irrelevant to the aims of the study, I suggest deleting or considerably reducing this paragraph.

Additional comments

This manuscript is a very interesting one, incorporating established methodology and previously verified concepts, using them in a novel approach and drawing important and useful conclusions. The text is well written, aims are clear, concise and the ideas well expressed. The definitions are precise, the methodology is clear and comprehensible. It will certainly be discussed as an important addition to our understanding of the most successful and diversified ‘placoderm’ group and generate new investigations. I suggest it to be published with minor revisions.

---

## Round 0.2 · accepted · Accept

Thank you for your close attention to the comments from the reviewers. I have assessed the changes made, and am satisfied with the current version. In my opinion, the manuscript is now ready to advance to publication.